# Linker-Tuning: Optimizing Continuous Prompts for Heterodimeric Protein Prediction

## Abstract

Predicting the structure of interacting chains is crucial for understanding biological systems and developing new drugs. Large-scale Pre-trained Protein Language models (PLMs), such as ESM-2, have shown an impressive ability to extract biologically meaningful representations for protein contact and structure prediction. In this paper, we show that ESMFold, which has been successful in computing accurate atomic structures for single-chain proteins, can be adapted to predict the heterodimer structures in a lightweight manner. We propose Linker-tuning, which learns a continuous prompt to connect the two chains in a dimer before running it as a single sequence in ESMFold. Experiment results show that our method is significantly better than the ESMFold-Linker baseline, with relative improvements of +28.13% and +54.55% in DockQ score on the i.i.d heterodimer test set and the out-of-distribution (OOD) test set HeteroTest2, respectively. Notably, on the antibody heavy chain light chain (VH-VL) test set, our method successfully predicts all the heavy chain light chain docking interfaces, with 46/68 medium-quality and 22/68 high-quality predictions, while being $9\times$ faster than AF-Multimer.

## 1 Introduction

Proteins are large biomolecules essential to life. They are sequences compromised of 20 types of amino acids and fold into three-dimensional (3D) structures to carry out functions. Predicting the 3D structures of proteins from amino acid sequences is a long-standing challenge in computational biology. It is important for the mechanical understanding of protein functions as well as for designing new drugs. In 2021, AlphaFold2 (AF2) strikes a huge success in solving this challenge, achieving near experimental accuracy on protein structure prediction [1]. However, this system heavily relies on Multiple Sequence Alignments (MSAs) to extract the evolutionary information, but MSAs are not always available or high quality, especially for orphan proteins and fast-evolving antibodies [2].

Inspired by the success of transformer language models in the field of Natural Language Processing (NLP), there is a line of work resorting to large-scale PLMs for protein structure prediction [2, 3, 4, 5]. These PLM-based models, such as ESMFold [3], take only amino acid sequences as input, eliminating the need for MSAs. Powered by PLMs, they show strong abilities in capturing protein structure information [6, 7]. And they are able to predict protein 3D structures at the atomic level with high accuracy while being an order of magnitude faster than AF2. However, these models are developed for predicting the structures of single-chain proteins and it is not clear how to use them to predict multi-chain protein structures.

To adapt these models for protein complex prediction, some researchers have proposed to use a poly-Glycine *linker* to join chains and input the linked sequences to the model to predict complex structures [8, 9]. The rationale is that the model should identify the linker segment as unstructured and fold the linked sequence in a similar way to multiple chains. Experimental result on AF2 shows

that this approach is simple yet effective. However, for the PLM-based models, whether a linker is effective or not for protein complex prediction remains unexplored. In the work of ESMFold, they briefly mention that they use a 25-residue poly-Glycine linker (denoted as G25 in the following) to join different chains for a specific protein complex example [3]. But they do not test the performance of the linker systematically. Based on existing work, we would like to investigate the following questions in this paper: 1) *How well can a G25 linker perform on protein complex prediction?* 2) *Can we optimize the linker to achieve a better result? And how?*

Viewing proteins as the language of life, linkers in fact are the same things as prompts in natural language. Inspired by prompt engineering [10, 11] in NLP, we propose Linker-tuning, which is to automatically learn a linker for the PLM-based model ESMFold on the task of heterodimeric protein structure prediction. Our goal is to find a linker that can link the two chains of a heterodimer so the structure prediction model can fold it in a similar way as a single-chain protein. How to best achieve this goal, however, is non-trivial and remains under-explored for the complicated protein structure prediction model. Through preliminary analysis, we find that it is better to place linker optimization at the Folding Module instead of at the PLM, which is different from intuition.

Considering ESMFold is a model with large-scale pre-trained PLM ESM2 that scales up to 15B parameters, to accelerate the linker learning procedure, we train and select our model on a proxy task called *distogram prediction* [12], a task that aims to predict inter-residue distance bins in the 3D space for each pair of residues in a given protein. After training, we test our learned linker on the 3D structure prediction task on three datasets to investigate the generalization ability of our method.

In summary, our main contributions are as follows:

- We propose Linker-tuning, a lightweight adaptation method that automatically learns a linker in the continuous space to adapt the single-chain ESMFold for heterodimer structure prediction.
- We show that our method outperforms the ESMFold-Linker baseline by large margins on both contact and structure prediction tasks on the heterodimer test set.
- We find that our method generalizes well to predict heterodimers with low sequence similarity and antibody VH-VL complex.

## 2   Biological background

**Linker**   In biology, linkers are short amino acid sequences created in nature to separate multiple domains in a single protein [13]. Biologists have found that linkers rich in Glycine act as independent units and do not affect the function of the individual proteins to which they attach [14, 15]. Therefore, we can use the Glycine-rich linker to join interacting chains to make it a single sequence, hoping it folds in the way they suppose to. Grounded in biological principles, we further extend the natural discrete linkers to virtual continuous linkers for better protein complex structure prediction.

**Distogram and contact map**   The 3D structure of a protein is expressed as $(x, y, z)$ coordinates of the residues' atoms in the form of a pdb file [16]. The distance between two residues in a protein 3D structure is defined as the Euclidean distance between their $C_\beta$ atoms ($C_\alpha$ for Glycine). Binning all the inter-residue distances in a protein into $k$ distance bins, we can obtain the distogram matrix [12]. For a protein with $L$ residues, the distogram $\boldsymbol{d}$ is an $L \times L$ matrix, with entry $\boldsymbol{d}_{ij}$ referring to the distance category of residue $i$ and $j$. In a coarser granularity, we can compute the contact map $\boldsymbol{c} \in \mathcal{R}^{L \times L}$, where $\boldsymbol{c}_{ij} = 1$ means the distance between residue $i$ and $j$ is less than or equal to 8Å. For protein complexes, we are especially interested in the inter-chain contact maps where the contacts are formed by two residues from different chains. The inter-chain contact map reflects the interface of interacting proteins, which is essential for predicting the 3D structure of the complex.

## 3   Related work

### 3.1   Protein structure prediction

**Single-chain protein structure prediction**   In recent years, single-chain protein structure prediction has attracted increasing attention from researchers in the Artificial Intelligence (AI) community,

mainly due to the ground-breaking success of the deep learning model AF2. Deep learning based protein structure prediction methods can be classified into two main categories: 1) MSA-based methods, such as AF2, that take protein sequences and MSAs as input and predict 3D structures [1, 17, 18]; 2) PLM-based methods, such as ESMFold, that take only protein sequences as input and predict 3D structures [3, 2, 4, 5, 19, 20, 21, 22, 23]. PLM-based methods do not rely on MSAs, which are time-consuming in searching homologs and not always available for some proteins like orphan proteins. Instead, they adopt large-scale pre-trained PLMs to learn evolutionary and structural meaningful representations for 3D structure prediction. In this work, we build our method upon PLM-based methods. Specifically, we adopt ESMFold [3] as the backbone since its code and pre-trained weights are all released and convenient to use. The overall architecture of ESMFold contains two parts: 1) *ESM2*: a PLM pre-trained with masked language modeling objective and scales up to 15B parameters; 2) *Folding Module*: contains Folding Trunk (similar to Evoformer in AF2) and Structure Module (same as the one in AF2), which are responsible for structure folding.

**Multi-chain protein structure prediction** In biology, multi-chain proteins are protein complexes formed by interacting single-chain proteins where the interactions are driven by the same physical forces as protein folding [24]. Recently, there is a line of work repurposing single-chain AF2 for protein complex structure prediction. The methods can be summarized into two main categories: 1) input-adapted methods that provide AF2 with pseudo-multimer inputs either by adding a large number to the residue_index between chains to indicate chain break [25, 26, 27, 28] or using a linker to join chains [8, 9]; and 2) training-adapted methods that retrain AF2 on multimeric proteins, such as AF-Multimer, the state-of-the-art (SOTA) method [29]. On the one hand, the two types of methods either do not update any parameters, or update all parameters of the base model, while our method falls in between, adding only a tiny number of extra parameters to the base model. On the other hand, existing work mainly focuses on the MSA-based method AF2, with little attention being paid to the PLM-based methods. In this work, we focus on adapting the PLM-based methods for two-chain protein structure prediction, which has not yet been explored.

## 3.2 Prompt engineering

In the NLP community, with the rise of large-scale pre-trained language models (LMs) such as GPT-3 [30], "pre-train, prompt, and predict" has become a prevalent paradigm to steer the LM to perform a wide range of downstream tasks [10]. In this paradigm, the downstream tasks are reformulated in a form that is similar to the LM pre-training task using a textual prompt [30, 31]. The key challenge in prompt-based learning is to find the right prompt for a specific task, termed "prompt engineering". There is a line of work that automatically search the right prompts for downstream tasks [32, 33]. In particular, instead of natural language prompts, some researchers propose to use continuous prompts, directly performing prompting in the embedding space of the LM [34, 11]. In their experiment, continuous prompts achieve strong results in both language understanding and generation tasks. In this work, we follow the idea of continuous prompting, searching for the linkers in the continuous space.

# 4 Method: Linker-tuning

To adapt the single-chain model for multi-chain protein structure prediction, we propose a lightweight adaptation method called Linker-tuning and a novel weighted distogram loss. The basic idea of our method is to optimize linkers, i.e., prompts, in the embedding space of ESMFold.

## 4.1 Problem formulation

Continuous linker tuning of ESMFold for protein complex structure prediction is a continuous optimization problem. Our goal is to find a linker that maximizes the performance of ESMFold on protein complex prediction. To be specific, we first denote training data as $D_{train} = \{(x_1, y_1), ..., (x_n, y_n)\}$ where $x_i = (x_i^A, x_i^B)$ and $x_i^A, x_i^B$ represent the amino acid sequences of two chains, $y_i$ is the structure of protein $x_i$. For a specified linker length $L$, the linker optimization problem is defined as follows:

$$\boldsymbol{l}^* = \operatorname*{arg\,min}_{\boldsymbol{l} \in E_L} \frac{1}{n} \sum_{i=1}^{n} \mathcal{L}(x_i, y_i, \boldsymbol{l}) \tag{1}$$

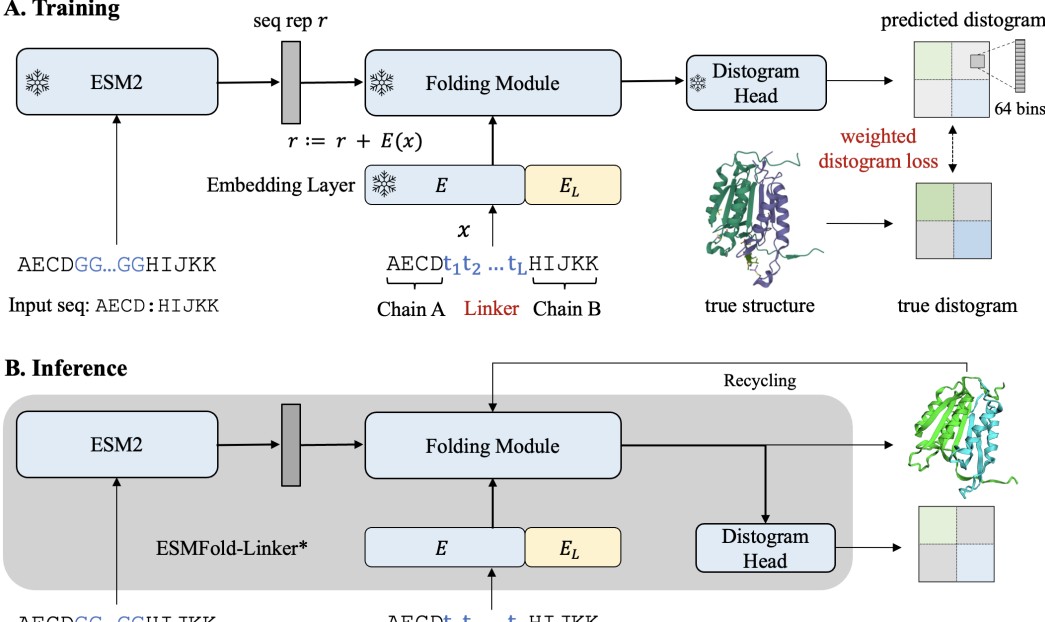

Figure 1: **Overview of Linker-tuning method with ESMFold as backbone**. **(A)** Training. Based on ESMFold (shown in blue colors), we add a linker embedding module $E_L$ (shown in yellow colors) with linker length $L$. Given a protein with multiple chains, we add the linker specified in the linker embedding module between each chain before running it as a single chain through the ESMFold model. The model outputs a distogram with the linker part removed. We use a weighted distogram loss as the objective function to train the linker embedding module while freezing all the parameters in ESMFold. **(B)** Inference. After training, ESMFold with our linker embedding module can be treated as a whole black box model, denoted as ESMFold-Linker*. The input for this model is just protein sequences. And the model outputs a predicted distogram as well as all the atoms' 3D coordinates for the protein.

where $l$ denotes a linker, $E_L \subset \mathcal{R}^{L \times d}$ denotes a specific embedding space with embedding dimension of $d$, $\mathcal{L}(x_i, y_i, l)$ denotes complex structure prediction loss w.r.t. protein $(x_i, y_i)$ using linker $l$. Therefore, the linker optimization is placed at the task level instead of at the instance level.

## 4.2 Model architecture

Our method is implemented based on ESMFold, a PLM-based strong structure prediction model. As shown in Figure 1, we place the continuous linker at Folding Module of ESMFold, which takes both the sequence representation from ESM2 and the amino acid sequence as input. There are two main reasons that motivate us to place the continuous linker at Folding Module instead of at ESM2. First, we can utilize the pre-trained distogram head while avoiding backpropagating to the giant ESM2 model. If we put it on the ESM2 side, the combined depth of training will go up to 104 layers, making it easily suffer from gradient vanishing and exploding. Second, preliminary analysis on inter-chain contact prediction (shown in Table 4) shows that using Folding Module on top of ESM2-3B increases prediction precision dramatically over ESM2-3B while ESM2-3B just performs slightly better ESM2-650M, implying that Folding Module is more sensitive to structure prediction and easier to control.

We implement a plug-in linker embedding module, which contains $L \times d$ learnable parameters where $d$ is the embedding dimension of Folding Module. During training, only the linker embedding module is trainable, while all the original parameters in ESMFold are frozen. Therefore, ESM2 is just a sequence feature extractor that generates features for Folding Module. As shown in Figure 1(A), we first use a poly-Glycine linker of the same length as the continuous linker to join different chains for the ESM2 input. Then we obtain the protein sequence representation and input it to Folding Module along with the chains connected by the continuous linker. Finally, the distogram head outputs a

probability distribution $p_{ij}^D \in \mathcal{R}^{64}$ of each residue pair $(i, j)$ on 64 distance bins, which is used for computing the loss function. After training, we view ESMFold and the linker embedding module as a whole and name it as ESMFold-Linker*. As shown in Figure 1(B), it can be used to predict the distograms as well as the 3D coordinates of all the residues for multi-chain protein sequences.

## 4.3 Weighted distogram loss

Intuitively, to predict the structure of a protein complex, we need to know two things: 1) the structures of each chain, on which ESMFold has been trained; and 2) the interaction interface between chains, which ESMFold has never seen before. Therefore, we propose to weight the intra-chain predictions and inter-chain predictions differently, with a focus on learning better interface between chains.

Formally, let $N_A, N_B$ be the number of residues in two chains in a protein complex, $N = N_A + N_B$ be the total number of residues in the protein complex. Let $y_{ij} \in \mathcal{R}^{64}$ denote the one-hot labels of the 3D space distance bins between residue pair $(i, j)$ and $p_{ij} \in \mathcal{R}^{64}$ be the corresponding predicted probability. We define a weighted distogram loss for a protein complex as follows:

$$\mathcal{L}(x, y, \boldsymbol{l}) = \mathcal{L}_1(x^A, y^A) + \mathcal{L}_1(x^B, y^B) + \lambda \mathcal{L}_2(x, y, \boldsymbol{l}) \tag{2}$$

where $\mathcal{L}_1(., .)$ denotes the single-chain distogram loss given as follows:

$$\mathcal{L}_1(x_A, y_A) = -\frac{2}{N_A(N_A + 1)} \sum_{i=1}^{N_A} \sum_{j \geq i}^{N_A} \sum_{b=1}^{64} y_{ijb} log(p_{ijb}^D) \tag{3}$$

and $\mathcal{L}_2(x, y, \boldsymbol{l})$ denotes the inter-chain distogram loss defined as follows:

$$\mathcal{L}_2(x, y, \boldsymbol{l}) = -\frac{1}{N_A N_B} \sum_{i=1}^{N_A} \sum_{j=1}^{N_B} \sum_{b=1}^{64} y_{ijb} log(p_{ijb}^D) \tag{4}$$

and $\lambda \geq 2$ is a hyperparameter controlling the attention we place on the interface of a protein complex. In our method, we use the weighted distogram loss as the training objective and validation metric.

# 5 Experiments

## 5.1 Experiment setting

**Datasets** We mainly perform experiments on heteromers of two chains. For training, we use the dataset from APOC [35], which contains heterodimers released in the Protein Data Bank (PDB) before 2018-09-30. After filtering out similar sequences at a 40% sequence identity threshold, it is split into train/valid/test[1] sets by CDPred [36]. We further filter out those proteins that contain missing $C_\beta$ coordinates ($C_\alpha$ for Glycine) in the pdb file. The resulting train/valid/test sample sizes are 2,946/193/172, respectively. The average number of residues in the test set is 367, with a maximum of 998. Furthermore, we use the largest blind test set HeteroTest2[2] from CDPred, which contains 55 heterodimers released in PDB between 2021-09-01 to 2021-10-20 [36]. The average number of residues is 505, with a maximum of 979. In addition, we use the antibody VH-VL test set from XtrimoDock [37]. It contains 68 samples released in PDB after 2022-02-01. Each sample consists of one heavy and one light chain, forming the fragment variable region (Fv), which is a critical part of antigen binding. The average number of residues is 231, with a range of [223, 244].

**Models** We use ESMFold-v1[3] as our backbone model. ESMFold-v1 consists of a 3B ESM2 model and a 670M Folding Module, which is the largest yet publicly available ESMFold checkpoint. For the Linker-tuning method, the linker length $L$ is set to 25, equal to the length of the manual poly-Glycine linker. So the plug-in linker embedding module contains 0.027M parameters. We initialize the linker embedding using the embedding of Glycine. During training, only the linker embedding module is trainable, while all the original parameters in ESMFold are frozen. The hyperparameter $\lambda$ in the

---

[1]https://github.com/BioinfoMachineLearning/CDPred/tree/main/example/training_datalists
[2]https://zenodo.org/record/6647564#.ZDWvMuxBxhE
[3]https://dl.fbaipublicfiles.com/fair-esm/models/esmfold_3B_v1.pt

weighted distogram loss is set to 4. We train the model on a single Nvidia A100 80GB GPU with batch_size=1 and num_epoch=15. The protein sequences in the training set are cropped to 225 residues to fit in GPU memory using the multi-chain cropping algorithm from AF-Multimer [29]. The number of recycles is set to 1 during training to reduce computation. We use Adam optimizer with a learning rate of 5e-4. We select the best model based on the validation weighted distogram loss. During inference, the number of recycles is set to 3.

**Baselines** We compare our method with several baselines and one SOTA model as follows:

- ESMFold-Linker: ESMFold-v1 with chains joined by the G25 linker as input.
- ESMFold-Gap: ESMFold-v1 with residue_index_offset set to 512.
- AlphaFold-Linker [29]: AF2 with a 21 residue repeated Glycine-Glycine-Serine linker.
- HDOCK [38]: rigid docking with single chains predicted by AF2.
- AF-Multimer(v3 best) [29]: AF-Multimer contains five models that are trained on all protein structures released in PDB before 2021-09-30. We take the best prediction from the five AF-Multimer models.

**Metrics** For protein complex 3D structure prediction, we use DockQ [39] to evaluate the quality of the predicted interfaces. As defined by Critical Assessment of PRediction Interactions (CAPRI), interfaces with $DockQ < 0.23$ means incorrect prediction, interfaces with $0.23 \leq DockQ < 0.49$ means acceptable prediction, $0.49 \leq DockQ < 0.80$ means medium quality prediction, and $DockQ \geq 0.80$ means high-quality prediction. To evaluate the whole predicted protein complex structure rather than the interfaces, we adopt two commonly used global structure metrics, namely, Root Mean Squared Deviation (RMSD), and Template-Modeling Score (TM-Score) [40]. Besides, we use the top-$k$ precision as an evaluation metric for inter-chain contact prediction. We set $k = N_s/5$, where $N_s$ is the minimum chain length for a given protein complex.

## 5.2 General heterodimer structure prediction

Table 1 shows the protein complex structure prediction results of our methods and the baselines on the heterodimer test set and HeteroTest2. On the i.i.d. heterodimer test set, ESMFold-Linker achieves a 0.32 DockQ score and a 0.76 TM-score on average. By optimizing the linker, our model, i.e., ESMFold-Linker*, achieves a 0.36 DockQ score and a 0.79 TM-score on average on the same test set, outperforming the ESMFold-Linker baseline by 13.61% and 3.28%, respectively. Interestingly, the gain of the interface quality (13.61%) is much larger than the gain of the whole structure quality (3.28%), indicating that our learned linker mainly improves the interfaces more than the overall structures. We further improve the ESMFold-Linker* by incorporating a large chain break, which adds a large number to the residue index in Folding Module. And the model ESMFold-Linker*-Gap achieves a 0.41 DockQ score and 0.80 TM-score, outperforming ESMFold-Linker by 28.13% and 5.26%, respectively. On the OOD test set HeteroTest2, we observe similar results. ESMFold-Linker*-Gap surpasses ESMFold-Linker by 54.55% DockQ score and 4.82% TM-score, respectively, suggesting that our learned linker can generalize well to OOD data.

Compared to AlphaFold-Linker, a model that takes linked sequences and MSAs as input, our best model ESMFold-Linker*-Gap achieves similar DockQ scores on both test sets, with lower values in RMSD. Meanwhile, it outperforms the classic docking method HDOCK with AF2 predicted chains as input in terms of DockQ score and RMSD. Furthermore, we compare it with the SOTA model AF-Multimer.[4] From Table 1, we can see there is still a large gap between our method and the AF-Multimer(v1 best) on HeteroTest2. There are three main reasons responsible for this gap: 1) The base model for AF-Multimer is AF2, which is a model stronger than ESMFold in general, especially for those proteins that have high-quality MSAs; 2) AF-Multimer is a fully fine-tuned version of AF2 on a larger protein complex structure dataset while our model is a prompt tuning method trained only on the heterodimer dataset; 3) AF-Multimer ensembles five models, while we only use one model. However, our method is able to predict some proteins that are hard for both ESMFold-Linker and AF-Multimer. As shown in Figure 2, ESMFold-Linker* successfully predicts the interface of the

---

[4]We use AF-Multimer v1 here because of the overlapping training data of AF-Multimer v3 and HeteroTest2. Since AF-Multimer v1 contains the heterodimer test set in its training data, we do not report the performance.

Table 1: Structure prediction results on **Heterodimer** data.

| | Heterodimer test | | | HeteroTest2 | | |
|---|---|---|---|---|---|---|
| | DockQ↑ | RMSD↓ | TM-score↑ | DockQ↑ | RMSD↓ | TM-score↑ |
| ESMFold-Linker | 0.32 ±0.34 | 10.76 ±8.68 | 0.76 ±0.19 | 0.11 ±0.20 | 20.10 ±10.31 | 0.62 ±0.19 |
| ESMFold-Gap | 0.34 ±0.35 | 10.37 ±8.89 | 0.77 ±0.19 | 0.11 ±0.21 | 20.17 ±11.70 | 0.63 ±0.19 |
| AlphaFold-Linker | **0.42** ±0.40 | 9.38 ±9.46 | **0.83** ±0.17 | 0.17 ±0.32 | 20.95 ±11.93 | 0.71 ±0.18 |
| HDOCK | 0.36 ±0.38 | 9.74 ±8.66 | 0.81 ±0.17 | 0.15 ±0.29 | 19.49 ±11.72 | 0.68 ±0.18 |
| ESMFold-Linker*(ours) | 0.36 ±0.35 | 9.19 ±8.04 | 0.79 ±0.19 | 0.14 ±0.23 | 19.03 ±10.93 | 0.65 ±0.20 |
| ESMFold-Linker*-Gap(ours) | 0.41 ±0.35 | **8.59** ±8.39 | 0.80 ±0.19 | 0.17 ±0.25 | 18.53 ±11.27 | 0.65 ±0.20 |
| AF-multimer(v1 best) | | | | **0.30** ±0.35 | **15.07** ±11.78 | **0.73** ±0.20 |

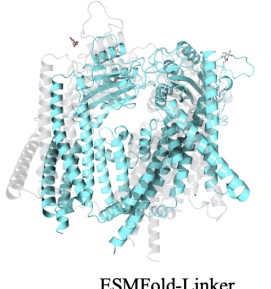
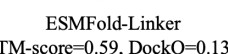
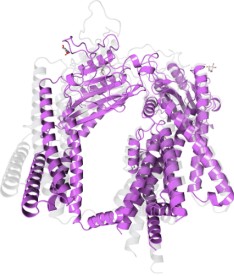
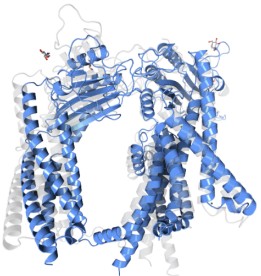

ESMFold-Linker
TM-score=0.59, DockQ=0.13

ESMFold-Linker*(ours)
TM-score=0.90, DockQ=0.39

AF-Multimer(v3 best)
TM-score=0.52, DockQ=0.02

Figure 2: Comparison of predicted structure quality and inference time of heterodimer **7D7F_AD** by ESMFold-Linker, ESMFold-Linker*(ours), and AF-Multimer(v3 best). 7D7F is a membrane protein comprising 917 residues in the A and D chains. Structures are drawn using Protein Imager [41]. Gray indicates the ground truth structure.

membrane protein 7D7F_AD with a DockQ score of 0.39 while ESMFold-Linker and AF-Multimer cannot predict the interface correctly.

## 5.3 Antibody heavy chain light chain docking

We further test our method on antibodies, an important type of protein in designing new drugs. Particularly, we focus on the heavy chain and the light chain docking. Table 2 shows the structure prediction results of MSA-free methods (first two methods) and MSA-based methods (last four methods) on the VH-VL test set. As an MSA-free model, ESMFold-Linker predicts all the interfaces successfully with an average DockQ score of 0.737, better than the classical docking method HDOCK. But it still lacks behind AlphaFold-Linker. For the three linker-based models, the distributions of their DockQ scores are shown in Figure 3. Equipped with the optimized linker, ESMFold-Linker* achieves an average DockQ score of 0.753, with 7 more high-quality interface predictions than ESMFold-Linker and 5 more high-quality interface predictions than AlphaFold-Linker. This result indicates that our learned linker trained on the general heterodimer dataset generalizes well to antibody data. Although the interface prediction performance of our method still falls behind AF-Multimer(v3 best), the gap in the DockQ score is much smaller compared to the case in HeteroTest2. Besides, it is quite close in TM-score to XtrimoDock [37], which is trained on an antibody-antigen dataset. Given our method only requires sequences as input, it can be a potentially useful model in the scenario of antibody design where the evolving antibody might not have MSAs.

Table 2: Structure prediction results on **VH-VL**.

| | DockQ↑ | RMSD↓ | TM-score↑ |
|---|---|---|---|
| ESMFold-Linker | 0.737
±0.084 | 1.459
±0.474 | 0.955
±0.019 |
| ESMFold-Linker*(ours) | 0.753
±0.083 | 1.388
±0.498 | 0.959
±0.019 |
| HDOCK | 0.705
±0.202 | 2.0318
±2.405 | 0.926
±0.101 |
| AlphaFold-Linker | 0.746
±0.089 | 1.4068
±0.520 | 0.957
±0.021 |
| AF-multimer (v3 best) | **0.779**
±0.091 | 1.287
±0.518 | 0.963
±0.020 |
| XtrimoDock | 0.775
±0.021 | **1.264**
±0.572 | **0.965**
±0.097 |

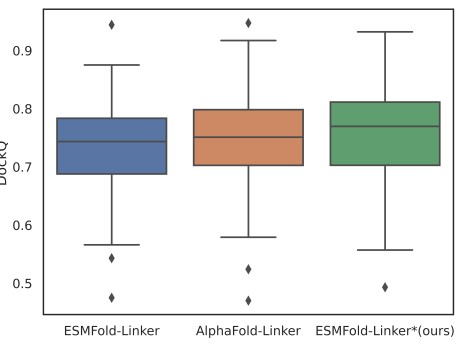

Figure 3: Boxplot of DockQ on **VH-VL**.

# 6   Analysis and Discussion

**ESMFold-Linker* is 9× faster than AF-Multimer in inference**
We report the structure inference time of the MSA-free methods (ESMFold-Gap, ESMFold-Linker, and ESMFold-Linker*) and the SOTA MSA-based model AF-Multimer on the VH-VL dataset using A100 80G GPU. Table 3 shows the total model inference time on the VH-VL test set, where AF-Multimer's time is only for one model, excluding the time of MSA search. As shown in Table 3, on the VH-VL test set with an average sequence length of 231, both ESMFold-Linker and ESMFold-Linker* take 4 minutes to run the inference, which is 9× faster than AF-Multimer.

Table 3: Inference time.

| | Time |
|---|---|
| ESMFold-Gap | 3 min |
| ESMFold-Linker | 4 min |
| ESMFold-Linker* | 4 min |
| AF-Multimer | 36 min |

**Large chain break or linker, or both?**    We perform an ablation study on ESMFold with chain break and linker to better understand the contribution of each operation. Table 4 shows the comparison of inter-chain contact prediction precision of ESMFold-based methods on the heterodimer test set and HeteroTest2.[5] As shown in Table 4, it is hard to tell whether ESMFold-Linker or ESMFold-Gap is better. However, combining the two (ESMFold-Linker-Gap) provides significant performance gains over using either operation alone on both datasets. We observe similar effects in our method when incorporating chain break with the optimized linker. Compared to using a chain break, the major limitation of using a linker is that it increases the computation cost (shown in Table 3). But we can enjoy the advantage of a large degree of freedom for improvement and better performance. Empirically, combining the two gives a better performance than just using each of them.

Table 4: Comparison of inter-chain contact prediction results on **Heterodimer** data.

| (%) | Heterodimer test | | | HeteroTest2 | | |
|---|---|---|---|---|---|---|
| | top Ns/5 | top Ns/2 | top Ns | top Ns/5 | top Ns/2 | top Ns |
| ESM2-650M-Linker | 12.02 | 9.89 | 8.33 | | | |
| ESM2-3B-Linker | 12.14 | 10.86 | 8.89 | | | |
| ESMFold-Linker | 49.88 | 47.04 | 40.64 | 23.00 | 18.92 | 13.72 |
| ESMFold-Gap | 51.15 | 48.13 | 40.82 | 22.09 | 18.21 | 13.08 |
| ESMFold-Linker*(ours) | 57.55 | 53.04 | 44.37 | 27.11 | 22.14 | 15.46 |
| ESMFold-Linker-Gap | 57.72 | 53.44 | 45.41 | 25.20 | 19.84 | 14.66 |
| ESMFold-Linker*-Gap(ours) | **60.40** | **56.27** | **48.00** | **28.00** | **23.69** | 17.26 |

**The learned linker allows more chain twist while rarely interacting with the chains**    In Figure 4, we visualize the predicted contact maps of two proteins with the linker inside to understand how the

---

[5]The contact map probabilities are obtained from the predicted distogram probabilities by summing the probability mass in each distribution below 8.25Å.

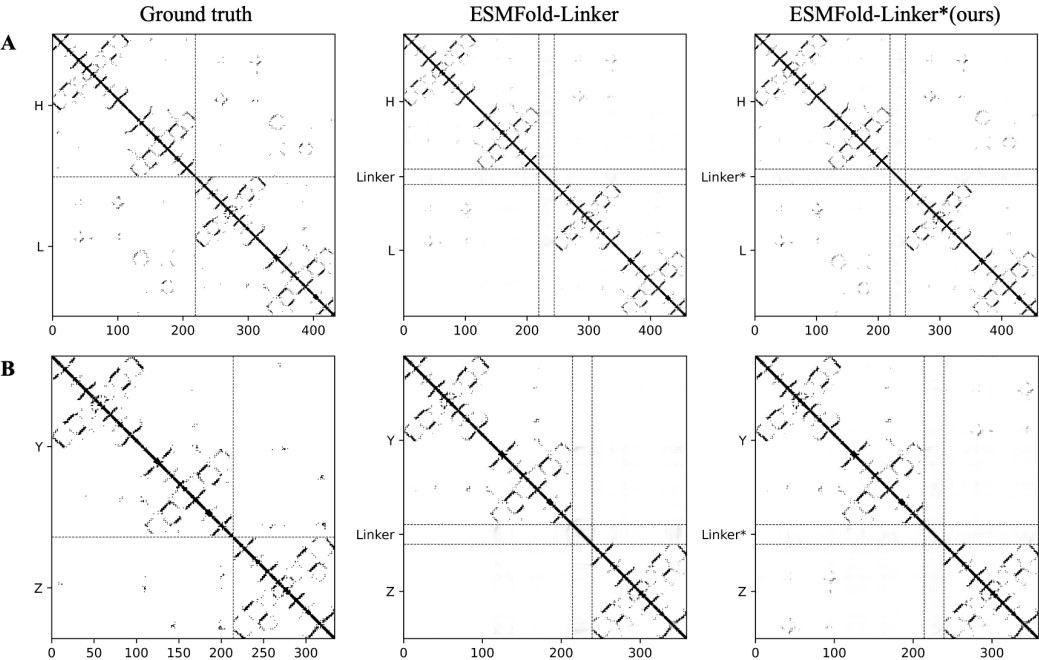

Figure 4: Contact maps of viral proteins **7VYR_HL (A)** and **7WPE_YZ (B)**.

linker interacts with the chains. The two proteins are 7VYR_HL and 7WPE_YZ, corresponding to a good case (0.77 DockQ score) and a bad case (0.01 DockQ score) in our model ESMFold-Linker*. As shown in Figure 4, both the G25 linker (middle) and our learned linker (right) seem to rarely interact with the protein chains in both cases. This result indicates that ESMFold is able to recognize the linker part as a disordered region and fold the connected sequences as multi-domain proteins. Furthermore, there are more predicted contacts using the learned linker than using the G25 linker in both cases. This result suggests that the learned linker allows the connecting chains to freely twist and rotate to recruit binding partners more than the manual linker.

**Limitations**    Our method has some limitations. First, if the base model (ESMFold-v1) is not good at predicting a certain type of protein complexes, such as the heterodimers in HeteroTest2, adding an optimized linker can not make it a strong model for that type of data since the trainable parameter size is very small. Second, our method is tested on heterodimers, whether it generalizes to homodimers or multi-chain proteins is unknown. Third, the linker is only optimized at the Folding Module, while the linker at ESM2 remains constant. And the linker length is treated as a hyperparameter, which can be further optimized to improve performance and speed.

# 7    Conclusions and future work

The use of prompts in protein structure prediction models is not always clear due to the high complexity of models and a general lack of biological knowledge for AI researchers. In this work, we have proposed Linker-tuning, a prompt tuning method to adapt the single-chain pre-trained ESMFold for heterodimer structure prediction. As proof-of-concept, we showcase that we can place a soft prompt in ESMFold. The task is reformulated as a pre-trained task itself under the biological prior. Experiments show that merely tuning a prompt on ESMFold can significantly improve the predicted complex structure quality over the discrete prompt handcrafted with strong biological insight. Hopefully, our work can inspire more work on AI for Protein Science.

There are two directions for future work. Firstly, we would like to extend our work to antibody-antigen structure prediction, a critical task with direct relevance to drug design. Secondly, we are going to explore structural-aware antibody design using our method since it is efficient and fast. By pursuing these directions, our objective is to make progressive contributions towards the development of effective drugs for disease treatment and pain relief.

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
