# OpenReview forum: "Linker-Tuning: Optimizing Continuous Prompts for Heterodimeric Protein Prediction"
_NeurIPS.cc/2023/Conference — Submitted to NeurIPS 2023_

### Official Review · Reviewer_Q87d · 2023-07-05

**Soundness:** 1 poor
**Presentation:** 3 good
**Contribution:** 2 fair
**Rating:** 3
**Confidence:** 5

**Summary:**

The paper proposes a strategy of predicting heterodimers with ESMFold.

Chains of heterodimers are linked by glycine linkers and inputed to esm. The output representations are then added with a learnable embedding layer, and then folded with the folding module in ESMFold. The finetuning is only done for the learnable embedding, with a weighted distogram loss.

Evaluations are done on 3 datasets, while one of them should be considered pointless. Some elevation is gained by the method, compared with directly using links.

**Strengths:**

1. The writing of the paper is clear.

2. The proposed method is reasonable and intuitive.

3. The method achieves comparable performances with other linker-based hacking strategies. Slight elevation of performances is gained (TMscore 0.62->0.65, ~0.03) from finetuning compared with directly using linkers.

**Weaknesses:**

Method:

3. The proposed idea, i.e. exploiting ESM models to predict protein complexes by architecture adjusting and finetuning has already been explored previously [1]. Seems that their implementations are more "neat": no external linkers are involved; permutation invariances are regarded; they solve complexes with arbitrary numbers of chains, both homomers and heteromers; and the performance elevation is more significant (TMscore 0.27->0.66, ~0.39 in their benchmark). However, no discussion let alone comparison is shown in this paper.

4. In fact, I don't know why linkers are needed to fold multimers at all: simply modifying the relative position indices suffices to tell the model that the residues are from separate chains. Involving linkers will implicitly pose in the model a geometrical constraint on the C-terminal of chain A and N-terminal of chain B (as they'll have relative position of +-L). Also the running time can be (slightly) added. Also, I don't personally like the saying that connects the linker idea to "prompts": they are totally different things. Doing so is more of attempting to ride the wave of LLM.

Evaluation:

5. The elevation of performances is in all sense too marginal. The results simply tell me that both ESMFold-linker and the proposed method are not reliable (avg DockQ of 0.11/0.17), instead of telling me that the proposed method is useful. In this case, maybe the percentage of success is a better metric to show.

6. The VH-VL docking benchmark is totally pointless and lacks commonsense. One who has basic senses of the domain knows that all protein folding efforts on antibodies should focus on Ab-Ag instead of VH-VL, because all interaction modes between VH-VL are the same, i.e. they fold almost identically (In table 2, as one can expect, all TM-scores are above 0.92). Therefore, they shouldn't be used as heterodimer folding or protein docking benchmarks. Even if one focuses on the structures of VH-VL, the metrics on the CDR loops should be independently reported, rather than global RMSD.

[1] Zhu et al, Uni-Fold MuSSe: De Novo Protein Complex Prediction with Protein Language Models. https://www.biorxiv.org/content/10.1101/2023.02.14.528571v1.full.pdf

**Questions:**

8. I don't know how the standard deviation of RMSD reachs 8.39 when the mean is 8.59 (0.35+- 0.36 for DockQ). This is so counter-intuitive. The authors may need a histogram to explain this.

9. Why is AF-multimer performances in Table 1 (left) missing?

10. Boxplots are shown only for VH-VL. Why not show them for Table 1 benchmarks?



**Limitations:**

limitations are addressed in section 6.

The suggestion would be

11. More solid benchmarks.

12. Discussions and comparisons with [1].

---

### Official Review · Reviewer_2TXp · 2023-07-06

**Soundness:** 3 good
**Presentation:** 3 good
**Contribution:** 2 fair
**Rating:** 4
**Confidence:** 4

**Summary:**

In this paper, the author applies prompt tuning in the heterodimeric protein prediction task. Instead of using the poly-Glycine linker, this method automatically finds the best linker in the continuous space. The author compares this method with several existing methods including the current start-of-art algorithm AF-multimer, the best PLM-based algorithm ESMFold-Linker, and the rigid docking algorithm HDOCK. The performance shows this method which is PLM-based is better than ESMFold-Linker but worse than AF-multimer.


**Strengths:**

1. The novelty and contribution of the paper mentioned in the paper are clear and correct.
2. The motivation behind the method and its design are well-founded and logical.
3. Overall, the writing is great.

**Weaknesses:**

1. Based on my understanding, the methodology seems to be limited in its applicability to the pre-trained language model-based protein structure prediction method, which is not considered the most accurate algorithm for protein structure prediction. Furthermore, the performance of this methodology appears to be influenced by the linker's position and the location suggested in the paper is not the PLM part, raising concerns about its compatibility with other PLM-based methods like Omegafold. Consequently, the paper's value may diminish if there are more robust folding algorithms available.
2. Given that all other methods are unsupervised, there is a possibility of the method benefiting from overfitting. Regarding the antibody dataset, from my understanding, antibodies generally exhibit a rigid overall structure except for the six CDR loops. Consequently, I suspect that the flexible CDR loop will not lead to significant variations in the docking of the light chain and heavy chain. Unless supported by evidence, I prefer to believe that the results obtained from the antibody dataset are not reliable and it may not be an appropriate dataset for this task. As for the Heterodimer test, while a 40% threshold seems acceptable, I believe setting a lower threshold, preferably below 30%, would be better.
3. In comparison to the state-of-the-art method AF-multimer, this method exhibits a significant decrease in performance. When people have the opportunity to utilize a substantial number of CPUs for preparing MSA information, the speed advantage offered by this method does not compensate for its accuracy limitations. Moreover, as far as I understand, there aren't a lot of downstream tasks that necessitate high-throughput calculations.

**Questions:**

1. The definition of the Gap model is not clear. It seems it is the version that you set the residue_index_offset for the second chain as 512. If the understanding is correct, is there any explanation for the improvement? It seems like this trivial strategy can achieve 50% of your improvement (according to the DockQ score for the Heterodimer test). Is that possible to also apply this trick to AF-multimer and AlphaFold-Linker?
2. The reason why you choose to use a 21-residue linker for AlphaFold-Linker which is different from the number you used for ESM.
3. Could you show the distribution of the score of the metrics for the Heterodimer test and HeteroTest2? It seems the standard deviation of scores is extremely high. In this case, it seems it is necessary to prove the difference is statistically significant for some key comparisons.

**Limitations:**

The limitation is well addressed and there is no potential negative societal impact of their work.

---

### Official Review · Reviewer_SavX · 2023-07-07

**Soundness:** 3 good
**Presentation:** 4 excellent
**Contribution:** 3 good
**Rating:** 5
**Confidence:** 5

**Summary:**

Inspired by the prompt tuning technique used in the field of NLP, the authors leverage the prompt tuning to adapt the single-chain pre-trained ESMFold for heterodimer protein structure prediction. To be specific, a learnable soft prompt is placed between protein chains. With such links, the pre-trained ESMFold treats complex structure prediction as the monomer structure prediction, and the prompt is tuned on the heterodimer dataset.

They show model with such a trick can outperform ESMFold-Linker baseline by large margins on both contact and structure prediction tasks on the heterodimer test set.

**Strengths:**

1 Leveraging the prompt tuning idea in multimer structure prediction using a single-chain pre-trained model is quite novel and interesting.

2. The proposed trick does improve the performance of the ESMFOLD-link baseline. The trick can potentially be applied to different single-chain pre-trained protein structure prediction models and can be considered a general trick.



**Weaknesses:**

1. The link-tuning idea is only validated on ESMFold. I'm curious about its effectiveness when applied to other protein structure prediction (PSP) models, e.g., Alphafold, and Omegafold. Prompt tuning is a general trick in NLP. Justifying the generalizability of the link-tuning trick on different PSP models can definitely make the manuscript stronger.

2. The idea of Linker-tuning is simple and effective, but I feel the current manuscript is not informative enough to be published on NeurIPS conference. If the authors can justify the generalizability of the link-tuning trick, I'm willing to adjust my score.

**Questions:**

NA

---

### Official Review · Reviewer_3KR1 · 2023-07-07

**Soundness:** 3 good
**Presentation:** 3 good
**Contribution:** 3 good
**Rating:** 4
**Confidence:** 4

**Summary:**

This work predicts the structure of heterodimeric protein chains by optimizing poly-G linkers that connect two chains of a heterodimer.

**Strengths:**

- This research, compared to other existing deep learning methods, finds an alternative to protein complex prediction methods, which is to make use of poly-G linkers. The idea connects closely to many biological applications and thus has more potential impacts on wider communities.
- The evaluation metrics cover various aspects to comprehensively assess the model's performance.

**Weaknesses:**

- ProteinMPNN as another famous multimeric structure prediction tool should have been compared.
- It seems very often the proposed method does not achieve the optimal performance (in Table 1 and Table 2).


**Questions:**

- It is not very clear to non-LM experts how adding a linker is an analogue to the prompt in LM.


**Limitations:**

No potential negative societal impacts were discussed in the main text.

---

> ### Author Rebuttal · Authors · 2023-08-06
>
> Thank you for your suggestions. We would like to clear up some misunderstandings and answer your questions.
> 1. About ProteinMPNN [1], it is a protein design model that takes structure as input and predicts amino acid sequence. It is not a structure prediction model. So we do not compare our model with ProteinMPNN.
> 1. About performance, our method is a very lightweight adaptation method built on ESMFold (with ESM2-3B). We cannot expect it to perform as well as the AF-Multimer for three reasons: (1) The base model for AF-Multimer is AF2, which is a model stronger than ESMFold (with ESM2-15B) in general, especially for those proteins with high-quality MSAs. (2) AF-Multimer is a fully fine-tuned version of AF2, with all the parameters in AF2 retrained, while our model is a prompt tuning method that contains only a tiny fraction of trainable parameters (0.0256M). (3) AF-Multimer ensembles five models, while we only use a single model.  Although the performance of our method is not as good as AF-Multimer, it is much faster and simpler both in training and inference. For a fair comparison, we think the baselines should be linker-based methods, including ESMFold-Linker and AF-Linker. As shown in Table 1 and Table 2, our method achieves better results than ESMFold-Linker. Furthermore, compared with AF-Linker, our method achieves comparable results on general heterodimers and better results on antibodies.
> 2. About the connection between linkers and prompts, let us take Natural Language Inference (NLI) task as an example. NLI aims to predict the relationship (entailed, contradicted, or neutral) between two given sentences. In the “Pretrain, Prompt, Predict” paradigm [2], the input can be reformulated as <Premise sentence> ? [MASK] <Hypothesis sentence>, where the prompt is represented as “? [MASK]”. The pretrained LM such as BERT will take the whole sentence as input and predict the masked token which will be further converted into one of the three answer choices. The key idea behind using prompts is to convert the downstream task into the pretrained task so it can be directly solved. This approach helps alleviate inconsistencies between the pretrained and downstream tasks and often leads to a better performance than the traditional fine-tuning method. In our task, we use the linker to convert the two chains into a single sequence  <chain 1> linker <chain2>. Then predicting a complex structure (downstream task) becomes the same as predicting a single-chain structure (pretrained task) under the consumption that the model will recognize the linker as an unstructured region. To sum up, linker and prompt are similar in three aspects: (1) they both serve as a connector, (2) they both convert the downstream task as the pretrained task, (3) they both steer the pretrained model to generate desired output by providing a specific context.
>
> Reference:
>
> [1] Dauparas, Justas, et al. "Robust deep learning–based protein sequence design using ProteinMPNN." *Science* 378.6615 (2022): 49-56.
>
> [2] Liu, Pengfei, et al. "Pre-train, prompt, and predict: A systematic survey of prompting methods in natural language processing." *ACM Computing Surveys* 55.9 (2023): 1-35.

---

> > ### Comment · Reviewer_3KR1 · 2023-08-17
> >
> > Thanks for the prompt response and for addressing the questions I raised. I understand the challenges inherent in expecting a lightweight model to match the performance of larger models. Nevertheless, the authors might consider incorporating a specific task or evaluating the model within a particular application context. By doing so, they can empirically demonstrate the indispensability of crafting a compact model, even if its performance may fall slightly short of optimal. This would greatly enhance the paper's impact and its recognition within the field. For the current version, I would like to maintain my original score.

---

### Decision · Program_Chairs · 2023-09-21

**Decision:**

Reject

**Comment:**

This paper studied the heterodimeric protein complex prediction, a fundamental task in biology. The author proposed to optimize a continuous prompt to connect the two chains before feeding the two sequences to the ESMFold protein structure prediction model.

The studied problem is definitely very important. Transferring the prompting tuning techniques widely used in NLP to the protein structure prediction context is also an interesting idea. However, there are some big concerns raised by the reviewers: (1) the generalization of the proposed approach. It seems that the proposed approach is only applicable for ESMFold but not for other protein structure prediction approaches such as AFMultimer; (2) the performance is not satisfying, still significantly lagging behind SOTA model such as AFMultimer. Due to the above limitations, the paper is not ready yet for publication at NeurIPS.